# Widespread Exaptation of L1 Transposons for Transcription Factor Binding in Breast Cancer

**DOI:** 10.3390/ijms22115625

**Published:** 2021-05-25

**Authors:** Jiayue-Clara Jiang, Joseph A. Rothnagel, Kyle R. Upton

**Affiliations:** School of Chemistry and Molecular Biosciences, The University of Queensland, St. Lucia, QLD 4072, Australia; jiayue.jiang@uq.net.au (J.-C.J.); j.rothnagel@uq.edu.au (J.A.R.)

**Keywords:** breast cancer, transposon, exaptation, transcription factor, transcriptional regulation, L1, LINE1, transcriptomics

## Abstract

L1 transposons occupy 17% of the human genome and are widely exapted for the regulation of human genes, particularly in breast cancer, where we have previously shown abundant cancer-specific transcription factor (TF) binding sites within the L1PA2 subfamily. In the current study, we performed a comprehensive analysis of TF binding activities in primate-specific L1 subfamilies and identified pervasive exaptation events amongst these evolutionarily related L1 transposons. By motif scanning, we predicted diverse and abundant TF binding potentials within the L1 transposons. We confirmed substantial TF binding activities in the L1 subfamilies using TF binding sites consolidated from an extensive collection of publicly available ChIP-seq datasets. Young L1 subfamilies (L1HS, L1PA2 and L1PA3) contributed abundant TF binding sites in MCF7 cells, primarily via their 5′ UTR. This is expected as the L1 5′ UTR hosts cis-regulatory elements that are crucial for L1 replication and mobilisation. Interestingly, the ancient L1 subfamilies, where 5′ truncation was common, displayed comparable TF binding capacity through their 3′ ends, suggesting an alternative exaptation mechanism in L1 transposons that was previously unnoticed. Overall, primate-specific L1 transposons were extensively exapted for TF binding in MCF7 breast cancer cells and are likely prominent genetic players modulating breast cancer transcriptional regulation.

## 1. Introduction

The normal growth and function of cells rely on the precise and coordinated expression of complex gene networks. Transcription is a primary step in gene expression and is regulated by functionally interacting transcription factors (TF), which recognise and bind specific cis-regulatory elements in genomic DNA [1,2,3]. Over 1600 TFs have been identified in the human genome through experimental evidence and computational prediction [4,5]. The regulatory impacts of human TFs are highly dynamic and associated with diverse cellular functions and disease phenotypes [5]. The specific interactions between TFs and their binding sites (TFBS) throughout the genome enable the coordinated regulation of gene networks, allowing functionally associated genes to be co-expressed [6,7,8].

Many human TFBSs are located within transposons, which in total occupy 45% of the human genome [9,10]. Transposons are repetitive DNA elements that can be hierarchically categorised into classes, families and subfamilies depending on their mode of replication, sequence structure, and evolutionary relatedness [9,11,12]. Some transposons, called the long terminal repeat (LTR) retrotransposons, originated from retroviral infections of the human cells, while the origin of the other human transposons remains largely unclear [9,13]. Despite their origins, transposons are often thought to be genomic parasites, and their ancestral sequences possess cis-regulatory elements that can be recognised by the human transcriptional machinery, such as RNA polymerases and various TFs [9,13,14,15,16]. These cis-regulatory elements allow the transposons to utilise the host’s transcriptional resources for their replication but also provide raw regulatory materials for the host to use for the transcriptional regulation of host genes.

Exaptation is the process where transposons contribute cis-regulatory elements for the regulation of host genes and has been described as a common phenomenon in various biological and disease contexts in the human genome [16,17,18]. Analysis of ChIP-seq data indicates that transposons contribute approximately 19% of all TFBSs within the human genome, including binding sites recognised by basic transcription regulators such as the TATA-box binding protein, as well as factors that mediate chromatin remodelling, such as CTCF and the chromodomain helicase DNA-binding proteins (CHD) [19]. Furthermore, transposons of the same families or subfamilies, which share a degree of sequence similarity, are often found to confer similar regulatory roles. For example, members of the MER41 subfamily are significantly enriched in binding sites for the immunity-associated IRF1 and STAT1 proteins and are found to contribute enhancer activity for genes involved in the immune signalling pathways, such as *AIM2* and *APOL1* [20,21]. In mammalian pregnancy, the MER20 transposons harbour binding sites for hormone-responsive and pregnancy-related TFs, such as C/EBPβ, PGR20 and FOXO1A [22]. Furthermore, retrotransposons in the LTR10 and MER61 families account for one-third of the tumour suppressor p53 binding sites, contributing to the global regulation of p53 target genes [23]. The genome-wide distribution of transposons makes them a suitable vector for dispersing regulatory sequences, particularly TFBS modules and thereby recruit functionally associated genes into co-regulation.

The most abundant class of transposons are the L1 transposons, with more than 500,000 copies colonising 17% of the human genome [9,24,25]. The majority of L1 transposons in the human genome arose prior to mammalian radiation [9,24,25]. The expansion of L1 transposons at different evolutionary time points has led to a diverse array of L1 subfamilies that can be further categorised by mutations in their sequences [24,26]. Despite the sequence variations, different L1 subfamilies share a common structure, where the intact L1 retrotransposon is about 6000 bp in length and contains a 5′ untranslated region (UTR), two opening reading frames (ORF) and a 3′ UTR ending in a polyadenylation signal and adenine homopolymer [9]. The 5′ UTR contains both a sense and an antisense promoter, allowing transcription to occur in both orientations [9,13,14,16,27].

L1 subfamilies that expanded during the primate radiation are denoted as “L1PA#”, where higher numbers indicate repeats of increasing age [25,26]. L1HS (or L1PA1) is the youngest L1 subfamily and contains L1 transposons that are unique to the human genome. The L1HS subfamily is the only actively transposing L1 subfamily in the human genome and contains approximately 100 copies of mobilisation-competent L1 transposons harbouring functional TFBSs for transcription [27,28]. The L1PA2 subfamily is approximately 4.5 million years older than L1HS and contains hominid-specific L1 transposons [26]. The L1 5′ UTR experiences a higher mutation rate than the other regions [29], and is prone to truncation, which is particularly common in the ancient L1 subfamilies [30,31] (Appendix A). Sequence analysis has suggested multiple origins of L1 5′ UTR sequences, which is hypothesised to lessen competition between co-existing L1 subfamilies for the host transcriptional resources [26].

Generally, human transposons, including L1 transposons, are heavily suppressed by epigenetic mechanisms and post-transcriptional silencing in normal somatic tissues [32]. The loss of epigenetic repression, such as in epithelial cancers, allows the exaptation of transposon-derived regulatory sequences [33,34,35,36]. There is growing evidence that L1 transposons contribute substantially to the transcriptional regulation of human genes, containing binding sites for a diverse range of TFs, such as CTCF, MAFK, RAD21 and SMC3 [19,37,38]. L1 transposons have also been found to contribute promoter activity to various genes. In particular, the antisense promoter located in the 5′ UTR (L1-ASP) leads to the production of chimeric transcripts composed of L1 sequences as well as neighbouring genomic sequences in various tissue types, including placenta, brain, and prostate [39,40]. Furthermore, extensive exaptation events of L1 transposons have been identified in cancer cells, likely due to loosened epigenetic repression and elevated transcriptional activation of L1 transposons in the cancer state [34,35,36]. In MCF7 cells, L1HS and L1PA2 transposons harbour binding sites for crucial regulators, such as C/EBPβ, E2F1, MYC and CTCF, with the TFBSs primarily clustering in the 5′ UTR [28,41,42]. The L1PA2 transposons are also found to drive alternative transcript expression in colorectal cancer and breast cancer, with the most notable example being the L1PA2-*MET* chimeric transcript, which is associated with enhanced malignancy [43,44,45,46]. In addition, other primate-specific L1 transposons have been found to drive pervasive transcription in cancer, giving rise to the aberrant expressions of genes including *SYT1* (L1PA2), *TYRP1* (L1PA3), *REL* (L1PA4) and *MRE11A* (L1PA5) [41,47].

We previously investigated the TF binding and promoter activity of an individual L1PA2 transposon in triple-negative breast cancer cell lines [41]. Upon further investigation, we demonstrate that TF binding in the context of breast cancer is a common feature within the L1PA2 subfamily, which contains approximately 5000 copies in the human genome [42]. These L1PA2 transposons contribute binding sites to critical TFs, including the estrogen receptor (ESR1), MYC and FOXA1. These TFs are associated with transcriptional misregulation in cancer, and their binding sites showed a pattern of co-localisation in L1PA2 transposons, confirming the role of L1PA2 transposons in dispersing TFBS modules and modulating co-operative TF binding [42]. The TF binding activity of L1PA2 transposons was specific to cancer and diminished in normal tissues, likely due to epigenetic repression in the normal somatic tissues [42].

Given the relatedness and structural similarities between L1 subfamilies, we hypothesised that the capacity for genomic regulation observed in the L1PA2 subfamily was a conserved feature amongst primate-specific L1 transposons and that sequence variations amongst L1 subfamilies, resulting from mutations and truncation of the L1 5′ UTR (Appendix A), would lead to diversity in the TF binding repertoire [30,31]. We thus aimed to investigate the landscape of transcriptional regulatory activity in primate-specific L1 subfamilies, ranging from the youngest L1HS subfamily to the more ancient L1PA12 subfamily. Using in silico approaches, we evaluated their regulatory potentials by predicting the putative TF binding profiles of various L1 subfamilies. Using cell-type-specific ChIP-seq data, we identified functional TFBSs within L1 transposons and confirmed the substantial contribution of primate-specific L1 transposons to TF binding in breast cancer. Young L1 subfamilies, including L1HS, L1PA2 and L1PA3, exhibited highly similar TF binding profiles and played a prominent role in contributing TFBS modules and thereby facilitating TFBS co-localisation. While the ancient L1 subfamilies often lacked intact 5′ UTR, they remained a rich reservoir of TFBSs, through alternative binding sites in the 3′ UTR. Furthermore, our results demonstrate that the cancer-specific regulatory activity was a conserved feature amongst the primate-specific L1 transposons. Together, these L1 transposons played a prominent role in mediating global TF binding activities in breast cancer.

## 2. Results

### 2.1. Primate-Specific L1 Subfamilies Are Predisposed to Diverse TF Binding

TFs exert regulatory roles by interacting with sequence features in genomic DNA or their binding motifs. Although putative binding motifs do not always define functional binding sites, they are indicative of potential TF binding activities. We scanned individual transposons of primate-specific L1 subfamilies for occurrences of known human TF binding motifs. A total of 114 out of 401 TFs from the HOCOMOCO core database were identified to have putative binding motifs in over 10% of transposons in each subfamily [48]. Hierarchical clustering revealed that when considering the entirety of transposon sequences, younger L1 subfamilies (L1HS-L1PA6) displayed a distinct putative TF motif profile, where a number of motifs appeared more prevalent in younger L1 subfamilies yet were identified at a lower frequency in the more ancient subfamilies (Figure 1A and Appendix A). These differences could potentially be attributed to multiple factors, including heavier sequence truncation in the ancient subfamilies (Appendix A), accumulated mutations, or result from selective pressures on successive L1 subfamilies. To estimate inter-subfamily variations in the TF binding profiles without the interference of sequence truncation and large insertions, we scanned only the intact 5′ UTRs (which contain the primary L1 sense and antisense promoters) of full-length transposons for known binding motifs and identified 63 human TFs with putative binding motifs in over 10% of 5′ UTRs. As found for all L1 sequences, distinctions between younger and more ancient L1 subfamilies were observed in the L1 5′ UTRs, confirming that the 5′ UTR sequence variations amongst L1 subfamilies likely predisposed L1 transposons to different TF binding activities (Figure 1B and Appendix A).

### 2.2. L1 Transposons Contain Abundant TFBSs in MCF7 Cells

Using the previously established “TEprofiler” pipeline and consolidated TFBSs from the GTRD and ChIP-Atlas databases [42,49,50], we investigated the TF binding activities in primate-specific L1 transposons in MCF7 breast cancer cells. All primate-specific L1 subfamilies analysed were identified to contain TFBSs in MCF7 cells, and the proportions of TF-bound L1 transposons ranged from 22.3% to 46% (Figure 2). The binding frequency increased to 81.82–99.71% when only intact L1 5′ UTRs were considered (Table 1). A total of 99 TFs were identified to bind at least one L1 transposon copy, while the specific binding frequencies differed amongst TFs and L1 subfamilies (Figure 2). Among these TFs, 89 TFs had at least one binding site in the L1 5′ UTRs. Considering all L1 transposons, ESR1 was the most frequently binding TF and were observed to bind all L1 subfamilies at a similar frequency, where an average of 76.52% of TF-bound L1 transposons contained at least one ESR1 binding site (Figure 2).

While ESR1 binding was highly prevalent in L1 transposons of various evolutionary ages, the youngest L1 subfamilies (L1HS, L1PA2 and L1PA3) displayed a distinct TF binding repertoire compared to the more ancient L1 subfamilies (L1PA4-L1PA12), where the youngest subfamilies were frequently bound by cancer-associated TFs, including MYC, FOXA1 and E2F1, as well as the nuclear receptor NR2F2 and chromatin remodelling-associated CTCF (Figure 2). These TFs were binding at a notably lower frequency in the ancient subfamilies (Figure 2). The substantial binding activities in young L1 subfamilies were associated with active epigenetic marks, particularly in their 5′ UTR, indicating transcriptional activation and potential promoter functions of these transposons (Appendix A).

The distinction in TF binding between young and more ancient L1 subfamilies was apparent when considering all L1 transposon copies in the human genome and could be attributed to varying structural integrity and sequence variations (Figure 3A and Appendix A). When considering only the intact L1 5′ UTRs, the differences in TF binding frequencies persisted, which confirmed that sequence variations in the L1 5′ UTRs predisposed L1 transposons to different TF binding potentials (Figure 3B and Appendix A). This unique TF binding pattern of young L1 subfamilies was also observed in other cancer cell lines, including K562 leukemia and A549 lung carcinoma cell lines, despite differences in TFs (Appendix A). Overall, the abundant and diverse TF binding profile of young L1 subfamilies was likely a result of a combination of structural integrity and the presence of unique sequence features.

### 2.3. 5′ Truncated L1 Transposons Exhibit Alternative Binding Sites

We next sought to investigate the distribution of TFBSs in different L1 subfamilies. As expected, TFBSs clustered in the 5′ UTRs of relatively young L1 subfamilies, and the TF-binding activity of the 5′ UTR was corroded by heavier 5′ truncation in the more ancient subfamilies. However, in the older L1 subfamilies, the 5′ UTR-lacking transposons contributed TF binding activities via their 3′ ends, which likely corresponded to their 3′ UTRs (Figure 4). Similar patterns were also observed when only ESR1 binding sites were considered, which explained the comparable levels of ESR1 binding across different L1 subfamilies despite an increasing degree of 5′ truncation in the ancient L1 transposons (Figure 2 and Figure 4).

### 2.4. TF Binding Motifs in L1 5′ UTRs Exhibit Sequence Variations

We next sought to explain the variability in TF binding observed in the intact L1 5′ UTRs. We hypothesised that the differential TF binding between young and more ancient L1 subfamilies was due to variations in the DNA sequence features that the TFs recognised and interacted with. To test that hypothesis, we scanned the L1 5′ UTRs for the known motifs of ESR1, which bound all L1 subfamilies at a similar frequency, as well as FOXA1 and E2F1, which frequently bound young L1 subfamilies but showed fewer binding activities in ancient L1 subfamilies. We used motif models from the JASPAR database [52], which identified binding sites more effectively in the younger L1 subfamilies, suggesting the presence of alternative binding motifs in the older subfamilies (Table 2). Nevertheless, several ancient L1 transposons with functional binding sites were identified to have a known motif.

To visualise differences between subfamilies at the locations of these known binding motifs, we used a multiple sequence alignment of the L1 subfamily consensus sequences. The consensus sequences were derived from full-length individual transposons in each subfamily [26]. Sequence features that are conserved in the majority of individual transposons will likely be represented in the consensus sequences, while features that are unique to individual elements within a subfamily would not be incorporated. We thus used the consensus sequences for comparing the commonality of TF motifs within each subfamily.

For ESR1, motif scanning in individual L1 5′ UTR identified several regions that could account for binding. These regions exhibited different degrees of sequence conservation amongst the L1 subfamilies but showed a general trend of increasing differences between subfamilies that were further apart in their evolutionary ages (Figure 5). For both FOXA1 and E2F1, only one region in the L1 5′ UTR was identified to contain TF motif sequences. Similar to ESR1, younger transposons showed a higher degree of sequence similarity to each other in the FOXA1 and E2F1 motif regions, and their consensus sequences contained the model motifs of these TFs, which could explain the widespread binding events within these subfamilies (Figure 5). In contrast, the same regions appeared to contain extra sequences in the more ancient subfamilies, which likely impeded TF binding activities in these subfamilies (Figure 5).

### 2.5. Co-Localisation of TF Binding Modules Is Common in Young L1 Subfamilies

The replicative nature of transposons has made them an exceptional vector for dispersing TFBSs of functionally associated TFs. We thus examined the pattern of TFBS co-localisation within the primate-specific L1 subfamilies. The degree of TFBS co-localisation was inversely correlated with the evolutionary age of L1 subfamilies, where binding site co-localisation was a common event in young L1 subfamilies, yet gradually diminished as the evolutionary age of L1 subfamilies increased (Figure 6). 

### 2.6. L1 Transposons Exhibit Cancer-Specific Exaptation for TF Binding

Transposons are often heavily suppressed by epigenetic mechanisms in normal somatic tissues, which is loosened in epithelial cancers, allowing the transcriptional activation and functional exaptation of transposons [32,34,35,36,53,54,55,56]. Increased TF binding activity was observed in L1PA2 transposons in MCF7 breast cancer cells, compared to the MCF10A near-normal cells [42]. The MCF10A cell line is a non-tumorigenic cell line commonly used to model normal breast epithelial cells [57,58]. In the current study, we further examined this cancer-specific pattern of exaptation in the other L1 subfamilies. A total of 12 TFs had been studied in both MCF7 and MCF10A cell lines based on data from the GTRD and ChIP-Atlas databases and thereby provided a basis for comparison [49,50]. When considering all 12 TFs, L1 transposons, regardless of the subfamily, showed notably elevated TF binding activities in MCF7 cells (Figure 7). A small proportion of L1 transposons contributed TF binding activities in both cell lines or contained MCF10A-specific TF binding activities (Figure 7).

We next examined each TF individually for their dependency on L1-derived TFBSs. We identified a number of TFs, including CTCF, ESR1, MYC and JUN, to bind L1 transposons almost exclusively in the MCF7 cells while showing minimal L1-binding activity in the MCF10A cells (Figure 7). This cancer-specificity was less striking for some other TFs, such as FOS, RUNX1, TEAD4 and TP53. Some of these TFs showed the reverse TF binding pattern, where more L1 transposons contributed TFBSs in the MCF10A cells (Figure 7). Regardless of the TFs, primate-specific L1 subfamilies appeared to show a highly consistent pattern of TF binding activity, where they either consistently contributed elevated TF binding activities in MCF7 cells or showed comparable levels in both cell lines, suggesting the conservation of temporal transcriptional activation amongst subfamilies (Figure 7).

## 3. Discussion

There is growing evidence supporting the widespread exaptation of transposons as a means of human gene regulation. Transposons are ubiquitous in the human genome and contain cis-regulatory elements that can be recognised by human transcriptional regulatory proteins [9,13,14,15,16]. The sequence similarity between transposons of related evolutionary lineages, and their ubiquitous distribution in the human genome, provide them with an inherent advantage in being exapted for mediating gene regulatory networks. This transposon-derived network regulation was initially proposed in 1971 by Britten and Davidson [59] and has been widely supported by evidence from various biological and disease contexts, including pregnancy, immunity and cancer [20,21,22,23,28].

L1 transposons are the most abundant family of human transposons, occupying 17% of the total human genomic sequences [9]. There are approximately 500,000 copies of L1 transposons embedded in the human genome [9], and they are identified and categorised into subfamilies based on their sequence features [26]. The full-length consensus structure of L1 transposons contains a 5′ UTR, harbouring abundant TFBSs and promoter activities, as well as two protein-coding ORFs and a 3′ UTR [9,13,14,16,27]. The majority of human L1 copies show 5′ truncation, which likely occurred during their replication process prior to insertion in the current genomic locations (Appendix A) [30,31]. L1 transposons that are present only in the primate lineages, denoted as “L1PA#”, arose between 3 to 100 million years ago [26]. Sequence analysis of the L1 5′ UTR revealed a discontinuous evolutionary history of L1, where subfamilies that co-existed and evolved independently likely recruited different 5′ UTRs, possibly as a mechanism to minimise competition for transcriptional resources [26]. The L1 5′ UTR has also been found to accumulate mutations at a faster rate compared to the other L1 structural components, further contributing to the sequence variations amongst L1 subfamilies [29].

Studies have shown extensive exaptation of L1 transposons for human genome regulation. They contribute abundant TF binding activities, primarily via their 5′ UTR, to a diverse range of TFs [19,28,42]. L1 transposons are found to drive the expression of various transcripts from the antisense promoter located within their 5′ UTR, and their promoter activities have been described in brain, prostate, and placenta tissues [39,40]. Furthermore, transcriptomic profiling has revealed abundant L1-derived transcripts in the context of cancer, and some of the L1-derived transcripts have been linked to malignancy [41,43,44,45,46,47].

With a focus on breast cancer, we developed the “TEprofiler” analysis pipeline for identifying cell type- or tissue-specific transcriptional activities within transposons [42]. Our analysis revealed the prominent role of L1PA2 transposons in breast cancer transcriptional regulation, where they contributed binding sites to TFs with a functional role in transcriptional misregulation in cancer. They facilitated the co-localisation of these TFBS modules and contributed to the combinatorial regulation of gene networks and the subsequent rewiring of the cancer transcriptome. The functional similarity within the L1PA2 subfamily was found to be correlated with sequence similarity. In the current study, we extended the TF binding profiling to the other primate-specific L1 subfamilies, which exhibited different degrees of sequence truncation and mutations, and showed that the cancer-specific regulatory activities of L1PA2 transposons in breast cancer were common to its evolutionary relatives. Our results demonstrate widespread exaptation of primate-specific L1 subfamilies for TF binding in the MCF7 breast cancer model.

In our analysis, we showed while sequence truncation was a common event observed in all primate-specific L1 subfamilies, younger subfamilies generally contained more full-length elements, which harboured intact 5′ UTRs (Appendix A). We predicted the putative TF binding profiles of primate-specific L1 subfamilies by performing a motif scanning analysis on individual transposons. The analysis revealed abundant TF binding potential of L1 transposons, where they harboured sequence features that could be recognised and potentially bound to by over 100 human TFs (Figure 1). We found that the sequence differences amongst L1 subfamilies, owing to both sequence truncation and sequence variations, predisposed them to different TF binding activities. When considering the entirety of L1 structures, putative TF binding motifs were less abundant in ancient L1 subfamilies, possibly due to the increased degree of sequence truncation (Figure 1A). Furthermore, when considering only the intact 5′ UTR of full-length L1 transposons, the younger L1 subfamilies (L1HS and L1PA2–L1PA6) continued to harbour abundant TF binding motifs (Figure 1B). The sequence features of younger L1 subfamilies, including the actively transposing L1HS transposons, likely contained more “raw” genetic elements that predisposed them to TF binding and thereby conferred higher exaptation potential [27,28].

The rapid developments in high-throughput, parallel sequencing technologies have enabled the identification of genome-wide cis-regulatory elements. ChIP-seq technology has been widely applied to identify global TF binding activities, and an extensive collection of ChIP-seq-identified TFBSs has been compiled and made available by public databases, such as the GTRD and ChIP-Atlas databases [49,50]. Using the publicly available TF binding data, we investigated the TF binding activities of L1 transposons and confirmed that many putative TF binding motifs translated into functional TFBSs in MCF7 breast cancer cells. Depending on the L1 subfamily, up to 46% of L1 transposons contributed TF binding activities in MCF7 cells (Figure 2). L1 elements with intact 5′ UTRs were notably more likely to be bound by TFs, as up to 99.71% of intact L1 5′ UTR harboured TFBSs in MCF7 cells (Table 1). This further confirmed previous observations of L1 5′ UTR being the primary reservoir of TFBSs [28,41,42].

In MCF7 cells, L1 transposons were found to be bound by a total of 99 TFs, including ESR1, MYC, FOXA1 and E2F1, which have well-established and critical roles in cell growth and division and have been linked to breast cancer development [60,61,62,63,64,65,66,67] (Figure 2). ESR1, or the estrogen receptor, is a major regulatory protein in breast cancer transcriptional regulation [63,68]. ESR1 dominated the L1-derived TF binding profiles, having binding sites in an average of 76.52% of TF-bound L1 transposons (Figure 2). The L1 5′ UTR was previously shown to be the major reservoir of binding sites and ESR1 binding motifs [42]. Surprisingly, the frequency of TF binding, particularly the frequency of L1 transposons harbouring ESR1 binding sites, appeared relatively uniform amongst L1 subfamilies and was seemingly unaffected by the increased 5′ truncation in the ancient L1 subfamilies (Figure 2).

Upon closer examination, it was discovered that the loss of 5′ UTR in the ancient L1 subfamilies was compensated by alternative binding sites in the 3′ end, possibly in the 3′ UTR (Figure 4). The sequence similarities amongst L1 transposons impeded de novo motif discovery, and motif scanning analysis using known TF motif models failed to detect putative TF binding motifs in the 3′ end-derived TFBSs (data not shown). This likely reflected a bias in the current TF binding models, which are incompatible for detecting L1-derived binding motifs, particularly in the L1 3′ UTRs. Nevertheless, in addition to the established regulatory potentials of L1 5′ UTRs, L1 transposons, particularly the ancient L1 subfamilies, appeared to contribute abundant TF binding activities through alternative exaptation of the 3′ end (Figure 4). The L1 3′ UTR, including the poly(A) tail, plays an essential role in L1 replication [69,70]. The L1 RNA has previously been reported to form secondary structures in the 3′ UTR, which also harbours the recognition site of the L1 ORF2 protein [71]. Although the RNA secondary structure of L1 3′ UTR does not explain the TF-binding capacity of the L1 double-stranded DNA, it may provide a basis for sequence features in this region that enable protein binding.

In contrast to ESR1, the other TFs, such as E2F1, MYC and FOXA1, showed a differential binding frequency between the young (L1HS, L1PA2-L1PA3) and the more ancient L1 subfamilies, as their binding activity was more prevalent in young L1 transposons (Figure 3). This distinctive TF binding profile of the youngest L1 subfamilies was also observed in other cancer types, despite differences in TFs (Appendix A). This distinction was apparent when considering the entirety of transposons and was preserved when only the intact L1 5′ UTRs were considered for TF binding, suggesting that it was likely associated with structural integrity as well as sequence variations within the TF binding motifs (Figure 3). To investigate the correlation between differential TF binding activities and sequence variations, we performed motif scanning analysis in the L1 5′ UTRs, and explored the sequence differences using their consensus sequences, which were derived from full-length individual transposons in each subfamily, and were thus representatives of the most overrepresented sequence features in the corresponding subfamily [26]. Indeed, motif scanning analysis identified intact TF binding motifs in the 5′ UTRs of young L1 subfamilies, which could account for the functional TF binding (Figure 5). At the same time, sequence differences in the motif-harbouring regions were observed in the consensus sequences of ancient L1 subfamilies, where the binding motifs of FOXA1 and E2F1 were disrupted by additional sequences (Figure 5). These sequence variations likely impeded TF recognition at these sites and thereby obstructed TF binding in the ancient L1 subfamilies (Figure 5). It is worth noting that the motif scanning analysis using known TF motif models demonstrated excellent performance in detecting binding motifs in younger subfamilies but failed to detect a statistically significant binding motif in a proportion of ancient L1 subfamilies (Table 2). There were potentially functional binding motifs present in these subfamilies that were not recognised by the current models, suggesting that alternative binding motifs might be present in different L1 subfamilies to enable exaptation for transcriptional regulation in breast cancer.

Cis-regulatory elements of functional association show a pattern of co-localisation in the genome, where their genomic locations are often found in close proximity to each other [72]. Interestingly, the most frequently binding TFs, including MYC, E2F1 and FOXA1, not only showed a preference for younger L1 subfamilies but were also binding at a comparable frequency to each other, suggesting a pattern of binding site co-localisation (Figure 2). We thus evaluated the occurrences of binding site co-localisation within each L1 subfamily and compared that to the remainder of the genome. We found an inverse correlation between the frequency of TFBS co-localisation and the evolutionary age of L1 subfamilies. TFBSs co-occurred more frequently in the youngest L1 subfamilies, namely L1HS, L1PA2 and L1PA3, but this pattern diminished as the evolutionary age of L1 subfamilies increased (Figure 6). In fact, TFBS co-localisation was depleted in the older L1 transposons, likely due to sequence variations that disrupted TF binding, as well as TFBSs lost to sequence truncation (Figure 6). Overall, young L1 subfamilies showed a prominent capacity of mediating binding site co-localisation and thereby facilitated the orchestrated regulation by functionally associated TFs.

Transcriptional regulation occurs in a tissue-specific and context-specific fashion, where the interactions between regulatory TFs and the genomic DNA may change drastically from one cell state to another. One mechanism for exerting the context-specificity of transcriptional regulation in diseases is differential TF binding, observed as loss or gains of TF binding activities [73,74]. We previously showed that TF binding activity of the L1PA2 subfamily was cancer-specific, as their exaptation for TF binding diminished drastically in the MCF10A near-normal cell line [42]. In the current study, we showed that this cancer-specific exaptation was a conserved feature amongst the primate-specific L1 subfamilies, likely due to common epigenetic repression mechanisms that they were subjected to in the normal cells (Figure 7). Overall, L1 transposons showed an elevated degree of exaptation in MCF7 cells, but this pattern was highly dependent on the specific TFs (Figure 7). We identified a group of TFs that showed L1-derived binding almost exclusively in MCF7 breast cancer cells, including ESR1, CTCF and MYC (Figure 7). This phenomenon of increased exaptation was likely correlated with the global epigenetic re-activation of L1 transposons in cancer and may lead to rewiring of the cancer transcriptome [42]. In contrast, some TFs, including FOS, RUNX1 and TEAD4, showed comparable levels of L1-derived binding in MCF7 and MCF10A cells. However, although some L1 transposons showed binding activities in both cell lines, these TFs appeared to bind different L1 copies in the two cell lines of interest, which could also lead to differential gene expression in the cancer state (Figure 7). Overall, L1 transposons mediated cancer-specific transcriptional regulation by contributing to differential, and in some cases, elevated TF binding.

In this study, we investigated the exaptation of primate-specific L1 subfamilies for TF binding events in a breast cancer model. Our analysis pipeline demonstrated broad utility and identified abundant TF binding activities in the L1 transposons. Despite having a shared origin, the remnants of L1 mobilisation in the human genome demonstrated great structural and sequence diversity, owing to 5′ truncation as well as sequence variations that resulted from ancestral recruitment of different 5′ UTRs and accumulated mutations. These sequence variations predisposed L1 transposons to diverse TF binding events, where L1 subfamilies of similar evolutionary age displayed resemblance in TF binding profiles. Younger L1 subfamilies harboured substantial and comparable TF binding activities in their 5′ UTRs and contributed to TFBS co-localisation and the combinatorial regulation by functionally interacting TFs. Interestingly, the TF binding capacity of ancient L1 subfamilies was not disadvantaged by 5′ truncation. Instead, these transposons contributed abundant TF binding activities via alternative binding sites in their 3′ ends, suggesting the presence of raw sequence materials in the L1 3′ UTR that could be exapted for TF binding over evolutionary time. Despite the structural differences, these L1 subfamilies showed highly consistent, cancer-specific TF binding activity, where they contributed to differential TF binding between the cancer cell model and the near-normal cells. Overall, L1 transposons exhibited widespread exaptation in breast cancer cells through more diverse mechanisms than previously anticipated. Given their abundance in the human genome and the conserved co-binding patterns of oncogenic TFs, these primate-specific L1 transposons likely play a prominent role in modulating global transcriptional regulation in cancer, leading to alternations in the cancer transcriptome.

## 4. Materials and Methods

### 4.1. Genomic Locations of Genetic Entities

Transposons and TFBSs that mapped to alternate chromosomes were excluded from all analyses in this study. All genomic locations were in hg38 unless otherwise stated.

### 4.2. Genomic Locations and Sequences of L1 Transposons

The genomic locations of primate-specific L1 transposons (L1HS and L1PA2 to L1PA12) were retrieved from the UCSC RepeatMasker table [51,75]. Full-length L1 transposons were defined by having a length of over 6000 bp, and their 5′ UTRs were defined as the first 1000 bp on the 5′ end. Strand-specific DNA sequences of individual L1 transposons or L1 5′ UTRs were extracted from the hg38 FASTA file for main chromosomes [9,76] using BedTools Getfasta (v2.25.0) (-s option) [77].

### 4.3. Cell Type-Specific TFBSs from the GTRD and ChIP-Atlas Databases

The GTRD and ChIP-Atlas databases contained a large collection of TFBSs generated by compiling publicly available ChIP-seq datasets [49,50]. TFBSs, defined as meta-clusters, were acquired from the GTRD database (version 19.10) [49], including information on the cell lines and tissue types from which the TFBSs were identified. Cell type-specific TFBSs were identified by filtering the GTRD dataset by the cell line names. Genomic locations of closely located TFBSs of the same TFs were merged using BedTools Merge (-d 300) [77]. Cell type-specific TFBSs (hg38) were also obtained from the ChIP-Atlas database [50]. For each TF, genomic locations of overlapping TFBSs were merged using BedTools Merge [77]. To restrict our analysis to TFs with established transcriptional roles, only TFBSs of TFs in the dbTF database were considered [78]. Finally, the binding sites from GTRD and ChIP-Atlas were combined to produce the final collection of TFBSs for each cell line, where overlapping TFBSs of the same TFs were merged using BedTools Merge [77]. In particular, the final TFBS collections for MCF7 and MCF10A cells contained binding sites of 212 and 40 TFs, respectively, including 12 TFs that were studied in both cell lines.

### 4.4. Epigenetic Profiling of L1 Subfamilies in MCF7 Cells

The L1HS, L1PA2 and L1PA3 transposons were investigated for their epigenetic states in MCF7 cells using publicly available ChIP-seq and DNAse-seq datasets (for methods and sources of data, see Jiang et al. (2021) [42]). Briefly, L1 transposons and their neighbouring 20 kb regions (bin = 100 bp) were profiled for DNAse sensitivity, as well as active (H3K27ac, H3K4me1, H3K4me3 and H3K36me3) and repressive (H3K9me3 and H3K27me3) histone tail modifications. For ChIP-seq datasets, RPKM values were normalised to the input control and averaged within the subfamily to produce the normalised RPKM values. For DNA-seq data, average RPKM values within each subfamily were calculated.

### 4.5. Motif Scanning Analysis of Known Human TFs in L1 Transposons

To predict the TF binding profiles of transposons, known TF binding motifs in the human core mononucleotide collection were acquired from the HOCOMOCO (version 11) database, which contains 401 motif models [48]. Individual L1 transposons were scanned for occurrences of binding motifs using FIMO v5.0.5 (-norc option), where statistical significance was defined by *p*-value < 1 × 10^−4^ [79]. For each TF motif, the frequency of motif occurrences was calculated as the number of L1 transposons containing a putative motif, divided by the total number of transposons in the corresponding subfamily. Hierarchical clustering was performed using the occurrence frequencies in R 4.0.0, using the pheatmap function (clustering_distance_rows/cols = “manhattan”, clustering_method = “complete” or “ward.D2”). Motif scanning analysis and hierarchical clustering were repeated for L1 5′ UTR sequences.

### 4.6. Identification of TFBSs Overlapping L1 Transposons and L1 5′ UTRs

For each cell line, TFBSs overlapping L1 transposons were identified using the “TEprofiler” pipeline [42]. Briefly, overlaps between the genomic locations of TFBSs and each L1 subfamily were identified using BedTools Intersect, where an overlap was called when at least half of a TFBS overlapped with a transposon, using the -f option [77]. For each subfamily, positions of TFBSs in the consensus sequence were calculated using the “RepStart” and “RepEnd” information from RepeatMasker [51]. The distribution of TFBSs in the consensus sequence (between 0 and 6500 bp) was visualised by plotting the middle of the adjusted positions in a histogram (bin = 5 bp). For each TF, the L1-derived binding frequency was calculated as the number of L1 transposons containing its binding sites divided by the total number of TF-bound transposons in the corresponding subfamily. Hierarchical clustering of primate-specific L1 subfamilies and TFs was performed using the binding frequencies, following the methods described above. TFBS identification and hierarchical clustering were repeated for the genomic locations of L1 5′ UTRs.

### 4.7. Identification of ESR1, FOXA1 and E2F1 Motifs in TF-Bound L1 5′ UTRs

L1 5′ UTRs bound by ESR1, FOXA1 and E2F1 were scanned for binding motifs. Briefly, TF motif models were acquired from the JASPAR 2020 database for ESR1 (MA0112.2), FOXA1 (MA0148.4) and E2F1 (MA0024.2) [52]. DNA sequences of TF-bound L1 5′ UTRs were extracted from the hg38 FASTA file for main chromosomes [9,76] using BedTools Getfasta (v2.25.0) (-s option) [77]. Motif occurrences in L1 5′ UTRs bound by the corresponding TF were identified using FIMO v5.0.5 (-norc option), with a statistical significance threshold of *p*-value < 1 × 10^−4^ [79].

### 4.8. Multiple Sequence Alignment of L1 Consensus Sequences

The full-length consensus sequences of primate-specific L1 subfamilies were acquired from Khan et al. [26]. Multiple sequence alignment was performed using MUSCLE (version 3.8.31) using default settings [80]. The consensus sequences were examined and compared manually in MEGA X (version 10.1.5) for the presence of ESR1, FOXA1 and E2F1 motifs, identified in individual L1 5′ UTRs [81,82].

### 4.9. Investigating the Degree of TFBS Co-Localisation in L1 Transposons

We next investigated the events of binding site co-localisation for 10 frequently binding TFs (ESR1, FOXA1, MYC, CTCF, NR2F2, E2F1, ZNF143, RELA, ZFX and STAT3) in individual L1 transposons using the “TEprofiler” pipeline [42]. Briefly, two binding sites were thought to co-localise if they were located within 500 bp from each other. For each TF-TF pair, the frequency of TFBS co-localisation in each L1 subfamily was compared to the remainder of the genome using a one-tailed proportion z-test. To account for asymmetrical co-localisation, for each pair of TFs, the co-localisation frequency was calculated with each TF as the denominator separately. To account for multiple testing, statistical significance was defined by a *p*-value < 0.0005 $(\frac{0.05}{10^{2}}$).

## 5. Conclusions

Exaptation of transposons for the regulation of human genomes has been described and studied in a diverse range of biological and disease contexts. In this study, we demonstrated that the L1 transposons from related evolutionary lineages were exapted to contribute substantial TF binding activities in a breast cancer model. Primate-specific L1 transposons contained abundant “raw” cis-regulatory sequences that predisposed them to TF binding and conferred functional TF binding events in MCF7 breast cancer cells. While intact L1 5′ UTRs remained a prominent reservoir of TF binding activities, 5′ truncated L1 transposons were found to contribute abundant TF binding activities through alternative binding sites in the 3′ ends. This suggests that the mechanisms of transposons exaptation are likely more dynamic than previously anticipated. Current methodology and bioinformatic resources available for investigating transposon-derived transcriptional regulation tend to focus more on young, structurally intact L1 transposons and will likely lead to underestimation of L1 regulatory potential. We find it peculiar that there is a large increase in the number of functional binding events within the youngest L1 subfamilies. This pattern suggests a potential selective advantage for pro-tumorigenic transcriptional events during recent L1 propagation, at least in this breast cancer model. Overall, our findings demonstrate a correlation between evolutionary relatedness and functional conservation in human transposons and reveal primate-specific L1 transposons as prominent genetic players in breast cancer transcriptional regulation.

## Figures and Tables

**Figure 1 ijms-22-05625-f001:**
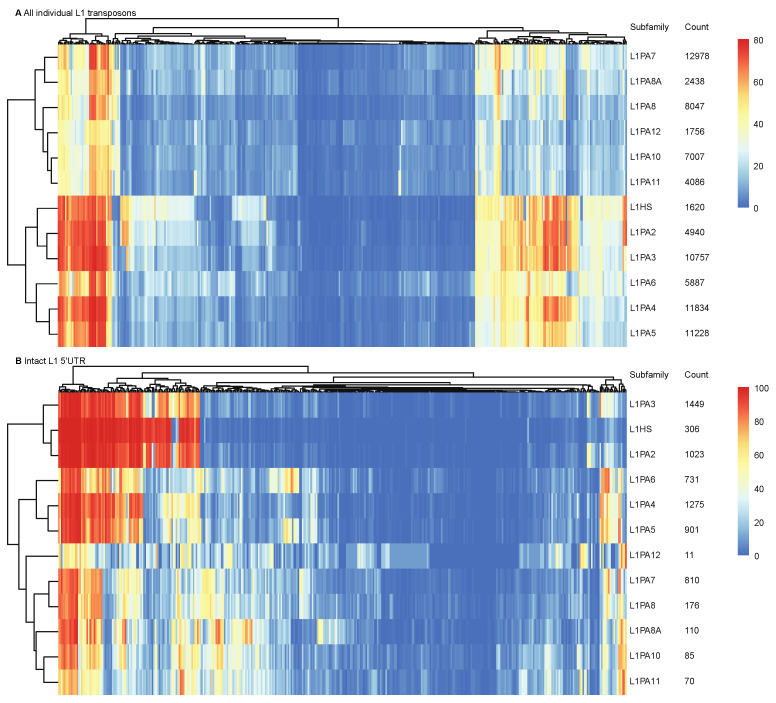
Different primate-specific L1 subfamilies were predisposed to different TF binding profiles. Motif scanning analysis was performed on (**A**) all individual transposons and (**B**) L1 5′ UTRs. Heatmaps show the hierarchical clustering performed on the occurrence frequencies of each TF motif (columns), using pheatmap (clustering_method = “complete”). The total count of L1 transposons and intact 5′ UTRs of each subfamily are shown.

**Figure 2 ijms-22-05625-f002:**
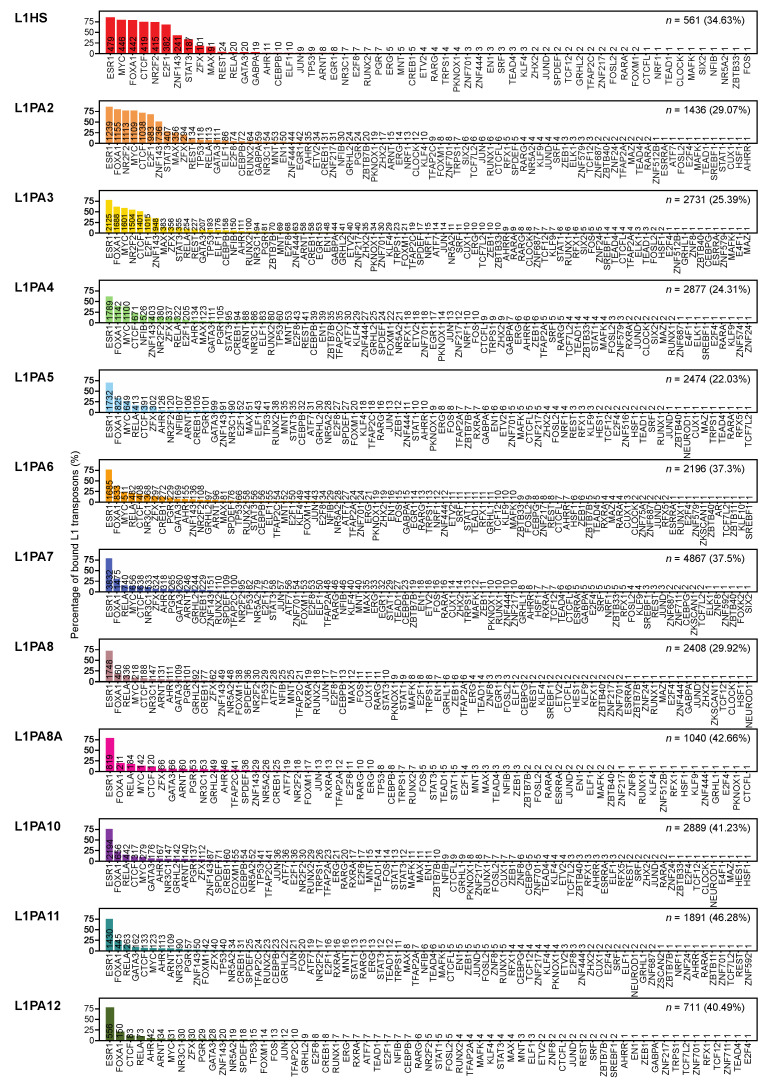
L1 transposon subfamilies harbour abundant TF binding activities in MCF7 cells. Bar graphs show the percentages of TF-bound L1 transposons harbouring TFBSs for each TF, labelled with the corresponding numbers of L1 transposons containing TFBSs. For each subfamily, the total TF-bound transposons are shown as the count (*n*) and the percentage of total L1 transposons in the subfamily.

**Figure 3 ijms-22-05625-f003:**
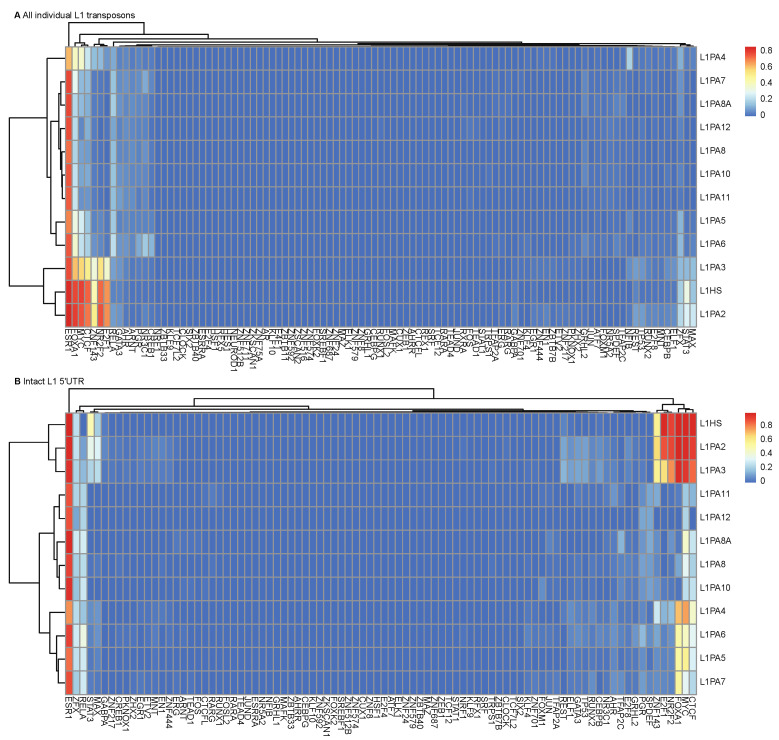
Young primate-specific L1 subfamilies displayed a distinct TF binding profile. TF binding activities in MCF7 cells were identified for (**A**) all individual transposons and (**B**) L1 5′ UTRs. Heatmaps show the hierarchical clustering performed on the TF binding frequencies, using pheatmap (clustering_method = “complete”).

**Figure 4 ijms-22-05625-f004:**
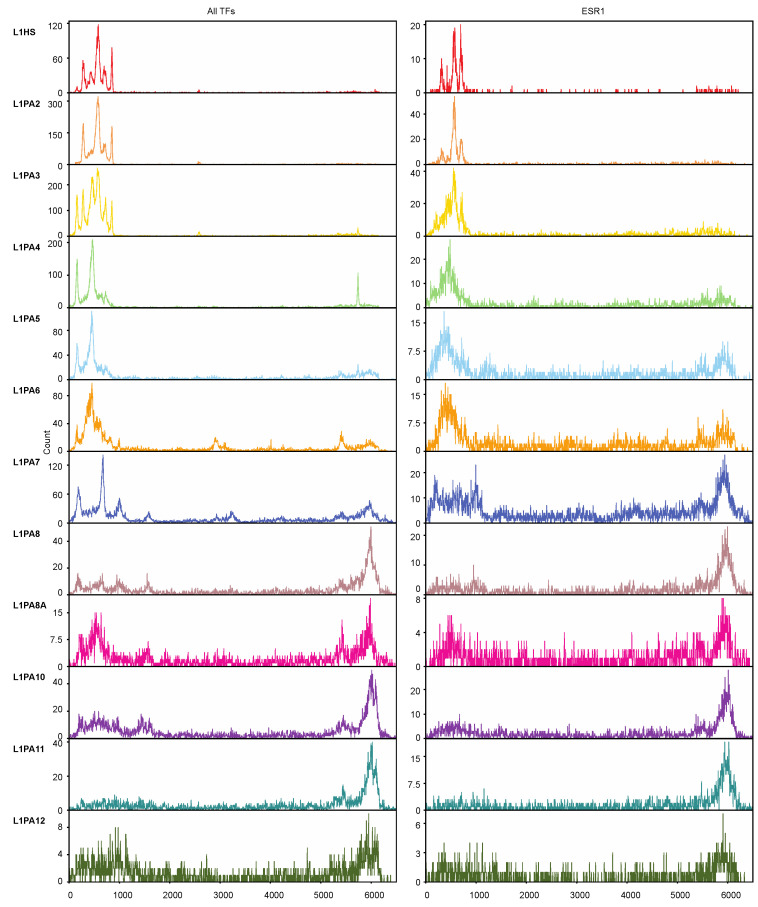
Truncated L1 transposons lacking 5′ UTRs contributed to TF binding through alternative binding sites in the 3′ ends. The histograms (bin = 5 bp) show the distribution of TFBSs of all TFs (left) and ESR1 (right) in the consensus sequences of each L1 subfamily (up to 6500 bp). The relative positions of TFBSs in the consensus sequences were calculated using the “RepStart” and “RepEnd” information from RepeatMasker [51].

**Figure 5 ijms-22-05625-f005:**
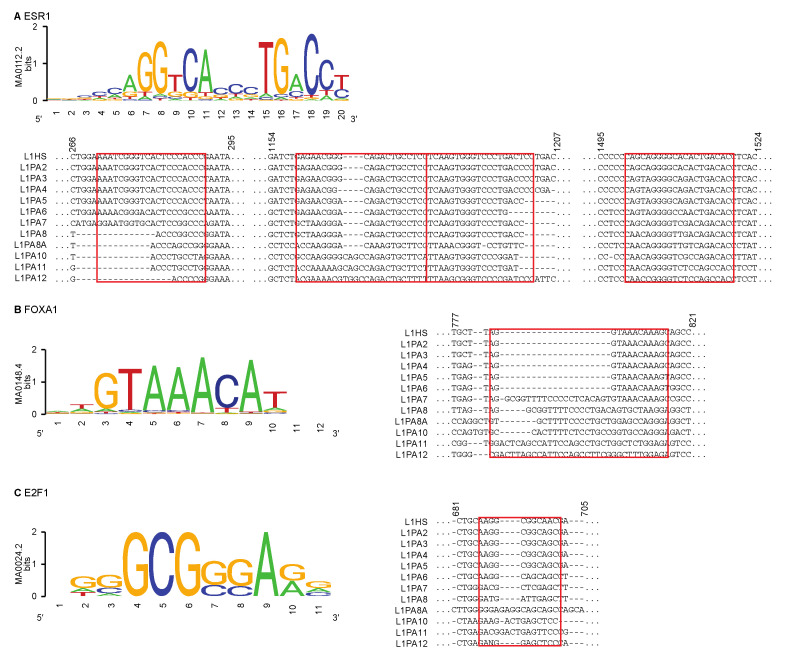
Differential TF binding activities correlated with sequence variations within the L1 5′ UTRs. For (**A**) ESR1, (**B**) FOXA1 and (**C**) E2F1, the motif models from the JASPAR database are shown [52]. Regions containing the motif sequences (highlighted in red rectangles) are shown in the multiple sequence alignment of L1 consensus sequences. The numbers correspond to the position of the bases in the multiple sequence alignment.

**Figure 6 ijms-22-05625-f006:**
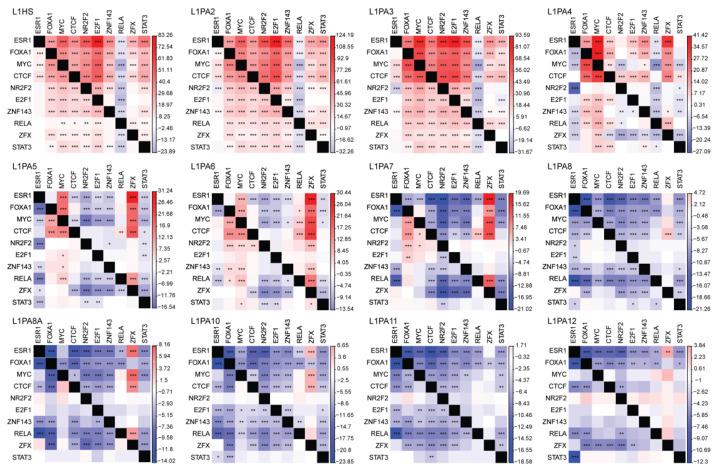
Young primate-specific L1 subfamilies contributed to TFBS co-localisation in MCF7 cells. The frequency of binding site co-localisation for frequently binding TFs was compared between L1 transposons and the rest of the genome (L1 versus non-L1) using a one-tailed proportion z-test. The heatmap indicates the z-scores of the proportion tests, where a positive z-score (red) indicates more co-localisation in L1, and a negative z-score (blue) indicates depletion of binding site co-localisation in L1. *p* < 0.0005: *; *p* < 0.0001: **; *p* < 0.00001: ***. The z-scores correspond to the frequency of the TF in the row co-localising with the TF in the column.

**Figure 7 ijms-22-05625-f007:**
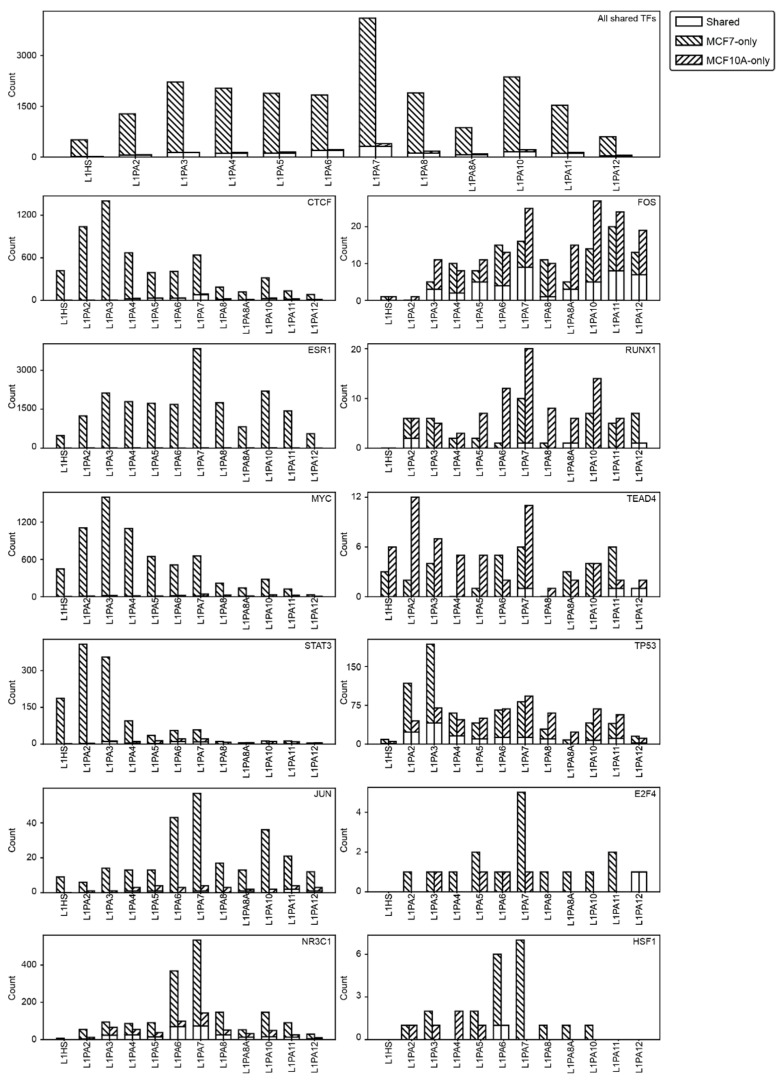
L1 transposons showed a consistent pattern of TF binding activities. Stacked bar graphs represent the number of L1 transposons contributed TFBSs in both MCF7 and MCF10A cells (empty bars), MCF7 cells only (downward stripes) and MCF10A cells only (upward stripes) are shown. The number of L1 transposons with TF binding activity are shown for all 12 TFs studied in both cell lines (top panel) and each individual TF (bottom panels).

**Table 1 ijms-22-05625-t001:** Number of total and TF-bound L1 5′ UTRs in MCF7 cells.

L1 Subfamily	Total Number of Intact 5′ UTR	Number (%) of 5′ UTRs with TFBSs
L1HS	306	305 (99.67%)
L1PA2	1023	1020 (99.71%)
L1PA3	1449	1434 (98.96%)
L1PA4	1275	1169 (91.69%)
L1PA5	901	801 (88.9%)
L1PA6	731	673 (92.07%)
L1PA7	810	742 (91.6%)
L1PA8	176	152 (86.36%)
L1PA8A	110	107 (97.27%)
L1PA10	85	80 (94.12%)
L1PA11	70	62 (88.57%)
L1PA12	11	9 (81.82%)

**Table 2 ijms-22-05625-t002:** The number of L1 5′ UTRs containing TFBSs in MCF7 cells and motifs of ESR1, FOXA1 and E2F1.

L1 Subfamily	ESR1		FOXA1		E2F1	
	Number Bound	Number with Motifs	Number Bound	Number with Motifs	Number Bound	Number with Motifs
L1HS	301	301	303	295	298	268
L1PA2	1002	998	1014	946	910	590
L1PA3	1369	1351	1398	1262	920	443
L1PA4	920	887	829	739	167	58
L1PA5	674	578	433	405	36	9
L1PA6	620	395	353	322	30	5
L1PA7	670	393	412	381	37	4
L1PA8	140	66	18	13	1	0
L1PA8A	105	40	3	1	1	0
L1PA10	77	30	1	0	3	0
L1PA11	55	20	2	0	0	0
L1PA12	8	3	0	0	0	0

## Data Availability

Data was obtained from the GTRD and ChIP-Atlas databases and are publicly available from http://gtrd.biouml.org/ and https://chip-atlas.org/, accessed on 3 May 2020 and 24 February 2021 respectively.

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
