# Peer review of "Widespread Exaptation of L1 Transposons for Transcription Factor Binding in Breast Cancer"

_ijms, 2021, doi:10.3390/ijms22115625_

Round 1

Reviewer 1 Report

<Comments to Authors>

The authors showed that primate-specific L1 transposones were exapted for TF binding in MCF7 cells. Although there were some points should be revised, entire of the study is interesting and worthy of eventual publication. The points are listed in below.

[Minor point]

  1. Please describe the clustering method of hierarchical cluster analysis in the manuscript. Although the default clustering method of pheatmap is “complete”, “ward.D2” is more better for analyzing omics data. If you used “complete” method, please explain me a rational reason.
  2. In figure 6, some characters were corrupted.
  3. Figure 7 was low resolution. Please adjust the resolution.

Author Response

The authors showed that primate-specific L1 transposons were exapted for TF binding in MCF7 cells. Although there were some points should be revised, entire of the study is interesting and worthy of eventual publication. The points are listed in below.

Comment 1

Please describe the clustering method of hierarchical cluster analysis in the manuscript. Although the default clustering method of pheatmap is “complete”, “ward.D2” is more better for analyzing omics data. If you used “complete” method, please explain me a rational reason.

Response to comment

We thank the reviewer for this note. We performed hierarchical clustering using different methods (by changing the “clustering_method” parameter in pheatmap). Our results for “ward.D2” and “complete” produced highly similar results with regards to the clustering of L1 subfamilies. To address this comment, we have added the ward.D2 clustering results to the manuscript (please see Supplementary Figure S2 and S4 for more information) and adjusted the Methods accordingly (please see track changes in line 572-573).

Comment 2

In figure 6, some characters were corrupted.

Response to comment

We thank the reviewer for pointing this out. We have double checked the texts in Figure 6 to make sure they were all visible. In addition, we increased the font size of the TF names to improve visibility.

Comment 3

Figure 7 was low resolution. Please adjust the resolution.

Response to comment

We thank the reviewer for the suggestion. We have replaced the original Figure 7 with a clearer image. In addition, we increased the spacing between the stripes in the bars to make them more visible.

Reviewer 2 Report

Review on the manuscript titled “Widespread exaptation of L1 transposons for transcription factor binding in breast cancer” by Jiang et al., 2021.

                The authors address the issue of impact of L1PA-mediated transcription factor binding sites (TFBS) in breast cancer originated MCSF7 cell line. Previously they have found abundant TFBSs in primate specific L1PA2 subfamily in cell lines (Jiang et al., 2021). In the presented study they performed extended in silico search for the several TFBS in primate-specific L1 subfamily, namely: L1HS, and L1PA2 to L1PA12. The authors analysed extensively TFBS for 3 specific TFs (SR1, FOXA1 and E2F), and performed TFBS co-location analysis for the extended set (SR1, FOXA1, MYC, CTCF, NR2F2, E2F1, ZNF143, RELA, ZFX and STAT3) followed by exhaustive TF list used in Fig 3. They used data from specialized TFBS databases underlining that many of the TFBSs are active (Chip-Seq data) specifically in L1PA subfamily MCF7 cell line compared to MCF10a (Fig.7).

                Finally, they concluded that “We find it peculiar that there is a large increase in the number of functional binding events within the youngest L1 subfamilies. This pattern suggests a potential selective advantage for pro-tumorigenic transcriptional events during recent L1 propagation, at least in this breast cancer model. Overall, our findings demonstrate a correlation between evolutionary relatedness and functional conservation in human transposons and reveal primate-specific L1 transposons as prominent genetic players in breast cancer transcriptional regulation”/

  • While the analysis performed shows thorough work done, there are several issues to raise.
  • The issue is interesting for two reasons: 1) L1PA ability to retropose recently, hence, they contain certain active TFBS for transcription; 2) they could be activated in certain cell lines as shown by authors. As a note, I state that L1 family is the one that extremely highly repressed since they represent the ‘enemy number one‘ for eukaryotic cells across Transposon spectra. Hence, it is extremely high chance that they are heavily methylated/heterochromized in the normal cells.
  • It’d be good to note in introduction whether and which TFBSs from L1 were exapted in human genome, at least the most abundant ones.
  • Why did authors omit L1PA15-16 from consideration?
  • While cell line used (MCSF7) is a model one for breast cancer, it may acquire quite specific chromatin state acutely different of newly arisen naive cancer cells.
  • MCF10A assigned as ‘near-normal cells’, the statement is referenced by authors previous publication (Jiang et al., 2021). Therein it is justified with L1 binding profile as well. The data independent of authors L1-related criteria should be used for proving it is ‘near normal’, since it is still cancer cell line that may maintain different evolution pathway than MCF7. If there are any aggressive scoring panels for comparing both, it should be provided, otherwise additional control should be used.
  • It is not clear how Fig3 a and b variants differ besides ranking/ordering of TFs. Is it meant to underline that all TFBS analysed are located in 5’UTR? Could be done in a less abundant manner? I’m aware that some TFBSs are located at 3’ UTR (Fig.4), but, judging by the topologies of Fig3 a/b these are dramatically low.
  • It looks like upward stripes vs downward stripes assignment should be reversed in Fig. 7 legend.
  • Chromatin state issue. I propose to authors scanning the chromatin HMM profile against L1PA mediated CTCF sites to see if they arrange insulator chromatin state (‘blue’, type 8) in any of the cell lines. That can support their function potential. If authors already did that, it would be quite relevant to mention.
  • English spellcheck should be performed since there are some instances of singular and plural nouns matching misspellings in particular.

The manuscript is well structured, written and represents the interest of cancer auditory.

Overall , the authors report valuable information on L1PA subfamily (in particular, L1HS, L1PA2, L1PA3) impact in cancer-related TFBS activity.  The mior point is that for final conclusion it'd be relevant finding other cell line with similar pattern for claiming ‘consistent pattern’ observation (not across TFBS, but cancer cell lines), otherwise it could be just one specific immortal cell line evolution pathway.  Gaining the insight on the putative  functional outcome of the TFs employed  would benefit the manuscript to my point.

Author Response

The authors address the issue of impact of L1PA-mediated transcription factor binding sites (TFBS) in breast cancer originated MCSF7 cell line. Previously they have found abundant TFBSs in primate specific L1PA2 subfamily in cell lines (Jiang et al., 2021). In the presented study they performed extended in silico search for the several TFBS in primate-specific L1 subfamily, namely: L1HS, and L1PA2 to L1PA12. The authors analysed extensively TFBS for 3 specific TFs (SR1, FOXA1 and E2F), and performed TFBS co-location analysis for the extended set (SR1, FOXA1, MYC, CTCF, NR2F2, E2F1, ZNF143, RELA, ZFX and STAT3) followed by exhaustive TF list used in Fig 3. They used data from specialized TFBS databases underlining that many of the TFBSs are active (Chip-Seq data) specifically in L1PA subfamily MCF7 cell line compared to MCF10a (Fig.7).

Finally, they concluded that “We find it peculiar that there is a large increase in the number of functional binding events within the youngest L1 subfamilies. This pattern suggests a potential selective advantage for pro-tumorigenic transcriptional events during recent L1 propagation, at least in this breast cancer model. Overall, our findings demonstrate a correlation between evolutionary relatedness and functional conservation in human transposons and reveal primate-specific L1 transposons as prominent genetic players in breast cancer transcriptional regulation”/

Comment 1

While the analysis performed shows thorough work done, there are several issues to raise.

The issue is interesting for two reasons: 1) L1PA ability to retropose recently, hence, they contain certain active TFBS for transcription; 2) they could be activated in certain cell lines as shown by authors. As a note, I state that L1 family is the one that extremely highly repressed since they represent the ‘enemy number one‘ for eukaryotic cells across Transposon spectra. Hence, it is extremely high chance that they are heavily methylated/heterochromized in the normal cells.

It’d be good to note in introduction whether and which TFBSs from L1 were exapted in human genome, at least the most abundant ones.

Response to comment

We thank the reviewer for the notes and suggestions. Firstly, we agree that L1HS, which is retrotransposition-competent, contain functional TFBSs that enable transcription. We have thus added this note to the Introduction and Discussion (please see track changes in line 85 and 394-395 respectively). Secondly, we agree that L1 transposons are heavily suppressed in normal tissues, possibly as a host’s mechanism to prevent transposition and unwanted gene expression. We have added information regarding the epigenetic state of transposons to the Introduction (please see track changes in line 92-95). Lastly, we further clarified the exaptation of L1 for TF binding by adding examples of TF binding to L1. In particular, we added:

“There is growing evidence that L1 transposons contribute substantially to the transcriptional regulation of human genes, containing binding sites for a diverse range of TFs, such as CTCF, MAFK, RAD21 and SMC3.” (please see track changes in line 97-98)

Comment 2

Why did authors omit L1PA15-16 from consideration?

Response to comment

We thank the reviewer for this question. While we tried to include as many primate-specific L1 subfamilies in our analysis as possible, our analysis was based on and also limited to the availability of data on the consensus sequences of each L1 subfamily and their genomic locations. The consensus sequences were acquired from Khan et al. (2006), and the genomic locations were acquired from the RepeatMasker database. We found that while the RepeatMasker recorded a subfamily of L1PA13, Khan et al. further classified these transposons into L1PA13A and L1PA13B. As we did not have the specific genomic locations of L1PA13A and L1PA13B transposons, we decided to limit our analyses to L1PA subfamilies younger than L1PA13 (thus excluding L1PA15 and L1PA16). We agree that TF profiling of the most ancient primate-specific L1 subfamilies will provide further insights into the exaptation mechanisms of L1s, and we intend to address this gap in knowledge in future studies.

Comment 3

While cell line used (MCSF7) is a model one for breast cancer, it may acquire quite specific chromatin state acutely different of newly arisen naive cancer cells.

Response to comment

We agree that the transcriptome and epigenome of MCF7 cells are likely different from those of the primary tumours. While primary tumours will provide invaluable insights into the molecular mechanisms of cancer development, it is technically challenging to perform a comprehensive analysis of the transcriptome and epigenome of primary tumours using publicly available data. This is due to the fact that fewer transcription factors and epigenetic marks have been profiled using primary tumours, as well as the heterogeneity within and across the diverse subtypes of breast cancer. Therefore, we chose to base our analysis on MCF7 cells, one of the most commonly used models for breast cancer. Nevertheless, our analysis pipeline has demonstrated broad utility in different cell lines, and can potentially be applied to study the transcriptome and epigenome of primary tumours.

Comment 4

MCF10A assigned as ‘near-normal cells’, the statement is referenced by authors previous publication (Jiang et al., 2021). Therein it is justified with L1 binding profile as well. The data independent of authors L1-related criteria should be used for proving it is ‘near normal’, since it is still cancer cell line that may maintain different evolution pathway than MCF7. If there are any aggressive scoring panels for comparing both, it should be provided, otherwise additional control should be used.

Response to comment

We thank the reviewer for this note. MCF10A is a commonly used immortalised cell line for modelling normal breast epithelial cells. To address this comment, we have provided a reference for the non-tumorigenicity of MCF10A cells in the manuscript:

“The MCF10A cell line is a non-tumorigenic cell line commonly used to model normal breast epithelial cells [57,58].” (please see tracked changed in line 306-307).

Comment 5

It is not clear how Fig3 a and b variants differ besides ranking/ordering of TFs. Is it meant to underline that all TFBS analysed are located in 5’UTR? Could be done in a less abundant manner? I’m aware that some TFBSs are located at 3’ UTR (Fig.4), but, judging by the topologies of Fig3 a/b these are dramatically low.

Response to comment

We thank the reviewer for the suggestions. Figure 3 shows the hierarchical clustering of primate-specific L1 subfamilies based on their TF binding frequencies. Figure 3A shows the clustering based all the TFBSs detected in all L1 transposons, including the full-length elements as well as the truncated elements. In contrast, Figure 3B shows the clustering based on the TF binding in only the 5¢ UTR of full-length elements. Both clustering results indicated that 1) the youngest L1 subfamilies (L1HS, L1PA2 and L1PA3) displayed highly similar TF binding profiles, and 2) their binding profiles were distinct from those of the more ancient L1 subfamilies. Figure 3A demonstrates that the TF binding profiles differed between the younger and older subfamilies, and that these differences could be due to two factors, namely varying structural integrity and sequence variations. By including Figure 3B, we re-examined the L1-derived TF binding profiles of only the 5¢ UTRs of full-length transposons, eliminating the factor of structural integrity. From Figure 3B, we concluded that when only structurally intact transposons were considered, the distinction between younger and older subfamilies persisted, which was likely due to sequence variations. To address this comment, we have further clarified our interpretation of Figure 3 in the manuscript:

“This distinction between young and more ancient L1 subfamilies was apparent when considering all L1 transposon copies in the human genome, and could be attributed to varying structural integrity and sequence variations (Figure 3A and Supplementary Figure S4). When considering only the intact L1 5¢ UTRs, the differences in TF binding frequencies persisted, which confirmed that sequence variations in the L1 5¢ UTRs predisposed L1 transposons to different TF binding potentials (Figure 3B and Supplementary Figure S4).” (please see track changes in line 207-212)

We decided to present the complete heatmaps and clustering results, as both provide direct visualisation of the TF binding profiles, while highlighting the differences between the younger and older subfamilies.

With regards to the 3¢ UTR TFBSs, we agree that the topology of Figure 3A and Figure 3B did not reflect the substantial TF binding activities in the 3¢ UTR of ancient L1 subfamilies. To address this concern, we would like to note that the heatmaps in Figure 3 showed the TF binding frequencies as percentages of TF-bound transposons/5¢ UTRs and are not intended to reflect of the total number of TF-bound transposons/5¢ UTRs (which were reported in Figure 2 and Table 1). The majority of ancient L1 transposons are 5¢ truncated; therefore, the total number of ancient L1s represented in Figure 3B are notably lower than that in Figure 3A. Taking ESR1 binding as an example for the purpose of demonstration, Figure 3A shows comparable binding frequencies across the L1 subfamilies, despite the increased 5¢ truncation in the ancient L1s. A small proportion of ESR1 binding in ancient L1s can be explained by binding sites in the 5¢ UTRs (Figure 3B), but the majority of ESR1 binding remained unaccounted for, which leads to the discovery of alternative TF binding sites in ancient L1 3¢  UTRs (Figure 4).

Comment 6

It looks like upward stripes vs downward stripes assignment should be reversed in Fig. 7 legend.

Response to comment

We thank the reviewer for pointing this out. We have changed figure 7 to make sure the figure legend is consistent with the patterns used.

Comment 7

Chromatin state issue. I propose to authors scanning the chromatin HMM profile against L1PA mediated CTCF sites to see if they arrange insulator chromatin state (‘blue’, type 8) in any of the cell lines. That can support their function potential. If authors already did that, it would be quite relevant to mention.

Response to comment

We thank the reviewer for this suggestion. We agree that epigenetic states of the transposons will provide invaluable information on their function potential. To address this gap in knowledge, we initially performed epigenetic profiling of the young L1 subfamilies (L1HS-L1PA3) in MCF7 cells, and have added the methods (please see section 4.4) and results to the manuscript (please see Supplementary Figure S3). The results demonstrate that L1 transposons were overall associated with active epigenetic marks, indicative of transcriptional activation and potential promoter activities. This observation has also been discussed in the manuscript:

“The substantial binding activities in young L1 subfamilies were associated with active epigenetic marks, particularly in their 5¢ UTR, indicating transcriptional activation and potential promoter functions of these transposons (Supplementary Figure S3).” (please see tracked changes in line 202-205).

Comment 8

English spellcheck should be performed since there are some instances of singular and plural nouns matching misspellings in particular.

Response to comment

We thank the reviewer for this suggestion. We have thoroughly checked the manuscript for spelling and made corrections where appropriate.

Comment 9

The manuscript is well structured, written and represents the interest of cancer auditory.

Overall, the authors report valuable information on L1PA subfamily (in particular, L1HS, L1PA2, L1PA3) impact in cancer-related TFBS activity.  The minor point is that for final conclusion it'd be relevant finding other cell line with similar pattern for claiming ‘consistent pattern’ observation (not across TFBS, but cancer cell lines), otherwise it could be just one specific immortal cell line evolution pathway. Gaining the insight on the putative functional outcome of the TFs employed would benefit the manuscript to my point.

Response to comment

We thank the reviewer for this suggestion. We agree that the L1-derived TF binding profiles in other cell lines will provide further insights. To address this comment, we performed TF profiling of L1 subfamilies in the K562 leukemia and A549 lung cancer cell lines, and added the results of hierarchical clustering to the manuscript (please see Supplementary Figure S5). In the manuscript, we highlighted the observation that the distinctive TF binding profile in young L1 subfamilies was also observed in the other cancer cell lines, despite differences in TFs (please see tracked changes in line 212-214 and line 440-441).